# Align before Fuse: Vision and Language Representation Learning with Momentum Distillation

**Junnan Li, Ramprasaath R. Selvaraju, Akhilesh D. Gotmare**
**Shafiq Joty, Caiming Xiong, Steven C.H. Hoi**
Salesforce Research
{junnan.li,rselvaraju,akhilesh.gotmare,sjoty,shoi}@salesforce.com

## Abstract

Large-scale vision and language representation learning has shown promising improvements on various vision-language tasks. Most existing methods employ a transformer-based multimodal encoder to jointly model visual tokens (region-based image features) and word tokens. Because the visual tokens and word tokens are unaligned, it is challenging for the multimodal encoder to learn image-text interactions. In this paper, we introduce a contrastive loss to ALign the image and text representations BEfore Fusing (ALBEF) them through cross-modal attention, which enables more grounded vision and language representation learning. Unlike most existing methods, our method does not require bounding box annotations nor high-resolution images. To improve learning from noisy web data, we propose momentum distillation, a self-training method which learns from pseudo-targets produced by a momentum model. We provide a theoretical analysis of ALBEF from a mutual information maximization perspective, showing that different training tasks can be interpreted as different ways to generate views for an image-text pair. ALBEF achieves state-of-the-art performance on multiple downstream vision-language tasks. On image-text retrieval, ALBEF outperforms methods that are pre-trained on orders of magnitude larger datasets. On VQA and NLVR$^2$, ALBEF achieves absolute improvements of $2.37\%$ and $3.84\%$ compared to the state-of-the-art, while enjoying faster inference speed. Code and models are available at https://github.com/salesforce/ALBEF.

## 1 Introduction

Vision-and-Language Pre-training (VLP) aims to learn multimodal representations from large-scale image-text pairs that can improve downstream Vision-and-Language (V+L) tasks. Most existing VLP methods (*e.g.* LXMERT [1], UNITER [2], OSCAR [3]) rely on pre-trained object detectors to extract region-based image features, and employ a multimodal encoder to fuse the image features with word tokens. The multimodal encoder is trained to solve tasks that require joint understanding of image and text, such as masked language modeling (MLM) and image-text matching (ITM).

While effective, this VLP framework suffers from several key limitations: (1) The image features and the word token embeddings reside in their own spaces, which makes it challenging for the multimodal encoder to learn to model their interactions; (2) The object detector is both annotation-expensive and compute-expensive, because it requires bounding box annotations during pre-training, and high-resolution (*e.g.* 600×1000) images during inference; (3) The widely used image-text datasets [4, 5] are collected from the web and are inherently noisy, and existing pre-training objectives such as MLM may overfit to the noisy text and degrade the model's generalization performance.

We propose ALign BEfore Fuse (ALBEF), a new VLP framework to address these limitations. We first encode the image and text independently with a detector-free image encoder and a text encoder. Then we use a multimodal encoder to fuse the image features with the text features through cross-modal attention. We introduce an intermediate image-text contrastive (ITC) loss on representations from the unimodal encoders, which serves three purposes: (1) it aligns the image features and the text features, making it easier for the multimodal encoder to perform cross-modal learning; (2) it

35th Conference on Neural Information Processing Systems (NeurIPS 2021).

improves the unimodal encoders to better understand the semantic meaning of images and texts; (3) it learns a common low-dimensional space to embed images and texts, which enables the image-text matching objective to find more informative samples through our contrastive hard negative mining.

To improve learning under noisy supervision, we propose Momentum Distillation (MoD), a simple method which enables the model to leverage a larger uncurated web dataset. During training, we keep a momentum version of the model by taking the moving-average of its parameters, and use the momentum model to generate pseudo-targets as additional supervision. With MoD, the model is not penalized for producing other reasonable outputs that are different from the web annotation. We show that MoD not only improves pre-training, but also downstream tasks with clean annotations.

We provide theoretical justifications on ALBEF from the perspective of mutual information maximization. Specifically, we show that ITC and MLM maximize a lower bound on the mutual information between different views of an image-text pair, where the views are generated by taking partial information from each pair. From this perspective, our momentum distillation can be interpreted as generating new views with semantically similar samples. Therefore, ALBEF learns vision-language representations that are invariant to semantic-preserving transformations.

We demonstrate the effectiveness of ALBEF on various downstream V+L tasks including image-text retrieval, visual question answering, visual reasoning, visual entailment, and weakly-supervised visual grounding. ALBEF achieves substantial improvements over existing state-of-the-art methods. On image-text retrieval, it outperforms methods that are pre-trained on orders of magnitude larger datasets (CLIP [6] and ALIGN [7]). On VQA and NLVR$^2$, it achieves absolute improvements of 2.37% and 3.84% compared to the state-of-the-art method VILLA [8], while enjoying much faster inference speed. We also provide quantitative and qualitative analysis on ALBEF using Grad-CAM [9], which reveals its ability to perform accurate object, attribute and relationship grounding implicitly.

## 2  Related Work

### 2.1  Vision-Language Representation Learning

Most existing work on vision-language representation learning fall into two categories. The first category focuses on modelling the interactions between image and text features with transformer-based multimodal encoders [10, 11, 12, 13, 1, 14, 15, 2, 3, 16, 8, 17, 18]. Methods in this category achieve superior performance on downstream V+L tasks that require complex reasoning over image and text (*e.g.* NLVR$^2$ [19], VQA [20]), but most of them require high-resolution input images and pre-trained object detectors. A recent method [21] improves inference speed by removing the object detector, but results in lower performance. The second category focuses on learning separate unimodal encoders for image and text [22, 23, 6, 7]. The recent CLIP [6] and ALIGN [7] perform pre-training on massive noisy web data using a contrastive loss, one of the most effective loss for representation learning [24, 25, 26, 27]. They achieve remarkable performance on image-text retrieval tasks, but lack the ability to model more complex interactions between image and text for other V+L tasks [21].

ALBEF unifies the two categories, leading to strong unimodal and multimodal representations with superior performance on both retrieval and reasoning tasks. Furthermore, ALBEF does not require object detectors, a major computation bottleneck for many existing methods [1, 2, 3, 8, 17].

### 2.2  Knowledge Distillation

Knowledge distillation [28] aims to improve a student model's performance by distilling knowledge from a teacher model, usually through matching the student's prediction with the teacher's. While most methods focus on distilling knowledge from a pre-trained teacher model [28, 29, 30, 31, 32], online distillation [33, 34] simultaneously trains multiple models and use their ensemble as the teacher. Our momentum distillation can be interpreted as a form of online self-distillation, where a temporal ensemble of the student model is used as the teacher. Similar ideas have been explored in semi-supervised learning [35], label noise learning [36], and very recently in contrastive learning [37]. Different from existing studies, we theoretically and experimentally show that momentum distillation is a generic learning algorithm that can improve the model's performance on many V+L tasks.

## 3  ALBEF Pre-training

In this section, we first introduce the model architecture (Section 3.1). Then we delineate the pre-training objectives (Section 3.2), followed by the proposed momentum distillation (Section 3.3). Lastly we describe the pre-training datasets (Section 3.4) and implementation details (Section 3.5).

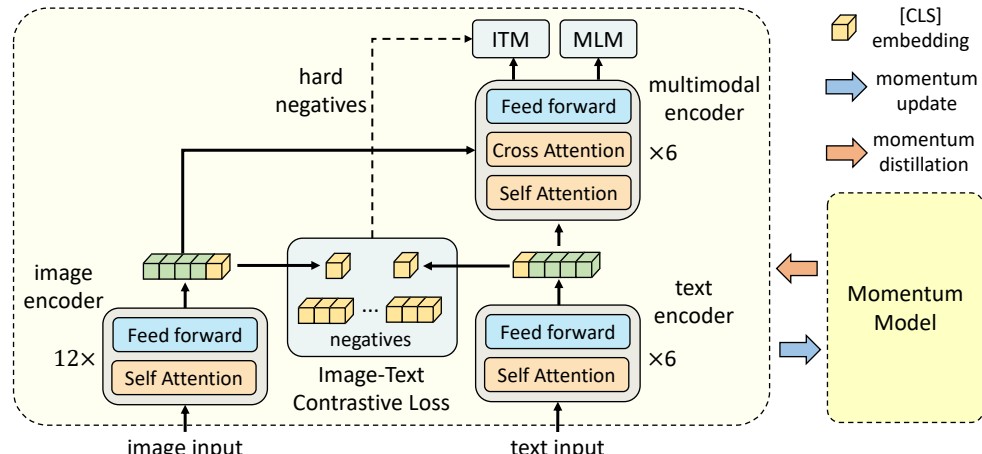

Figure 1: **Illustration of ALBEF.** It consists of an image encoder, a text encoder, and a multimodal encoder. We propose an image-text contrastive loss to align the unimodal representations of an image-text pair before fusion. An image-text matching loss (using in-batch hard negatives mined through contrastive similarity) and a masked-language-modeling loss are applied to learn multimodal interactions between image and text. In order to improve learning with noisy data, we generate pseudo-targets using the momentum model (a moving-average version of the base model) as additional supervision during training.

## 3.1 Model Architecture

As illustrated in Figure 1, ALBEF contains an image encoder, a text encoder, and a multimodal encoder. We use a 12-layer visual transformer ViT-B/16 [38] as the image encoder, and initialize it with weights pre-trained on ImageNet-1k from [31]. An input image $I$ is encoded into a sequence of embeddings: $\{\boldsymbol{v}_{\text{cls}}, \boldsymbol{v}_1, ..., \boldsymbol{v}_N\}$, where $v_{\text{cls}}$ is the embedding of the [CLS] token. We use a 6-layer transformer [39] for both the text encoder and the multimodal encoder. The text encoder is initialized using the first 6 layers of the BERT$_{\text{base}}$ [40] model, and the multimodal encoder is initialized using the last 6 layers of the BERT$_{\text{base}}$. The text encoder transforms an input text $T$ into a sequence of embeddings $\{\boldsymbol{w}_{\text{cls}}, \boldsymbol{w}_1, ..., \boldsymbol{w}_N\}$, which is fed to the multimodal encoder. The image features are fused with the text features through cross attention at each layer of the multimodal encoder.

## 3.2 Pre-training Objectives

We pre-train ALBEF with three objectives: image-text contrastive learning (ITC) on the unimodal encoders, masked language modeling (MLM) and image-text matching (ITM) on the multimodal encoder. We improve ITM with online contrastive hard negative mining.

**Image-Text Contrastive Learning** aims to learn better unimodal representations before fusion. It learns a similarity function $s = g_v(\boldsymbol{v}_{\text{cls}})^\top g_w(\boldsymbol{w}_{\text{cls}})$, such that parallel image-text pairs have higher similarity scores. $g_v$ and $g_w$ are linear transformations that map the [CLS] embeddings to normalized lower-dimensional (256-d) representations. Inspired by MoCo [24], we maintain two queues to store the most recent $M$ image-text representations from the momentum unimodal encoders. The normalized features from the momentum encoders are denoted as $g_v'(\boldsymbol{v}_{\text{cls}}')$ and $g_w'(\boldsymbol{w}_{\text{cls}}')$. We define $s(I, T) = g_v(\boldsymbol{v}_{\text{cls}})^\top g_w'(\boldsymbol{w}_{\text{cls}}')$ and $s(T, I) = g_w(\boldsymbol{w}_{\text{cls}})^\top g_v'(\boldsymbol{v}_{\text{cls}}')$.

For each image and text, we calculate the softmax-normalized image-to-text and text-to-image similarity as:

$$p_m^{\text{i2t}}(I) = \frac{\exp(s(I, T_m)/\tau)}{\sum_{m=1}^M \exp(s(I, T_m)/\tau)}, \quad p_m^{\text{t2i}}(T) = \frac{\exp(s(T, I_m)/\tau)}{\sum_{m=1}^M \exp(s(T, I_m)/\tau)} \tag{1}$$

where $\tau$ is a learnable temperature parameter. Let $\boldsymbol{y}^{\text{i2t}}(I)$ and $\boldsymbol{y}^{\text{t2i}}(T)$ denote the ground-truth one-hot similarity, where negative pairs have a probability of 0 and the positive pair has a probability of 1. The image-text contrastive loss is defined as the cross-entropy H between $\boldsymbol{p}$ and $\boldsymbol{y}$:

$$\mathcal{L}_{\text{itc}} = \frac{1}{2} \mathbb{E}_{(I,T)\sim D} \left[ \text{H}(\boldsymbol{y}^{\text{i2t}}(I), \boldsymbol{p}^{\text{i2t}}(I)) + \text{H}(\boldsymbol{y}^{\text{t2i}}(T), \boldsymbol{p}^{\text{t2i}}(T)) \right] \tag{2}$$

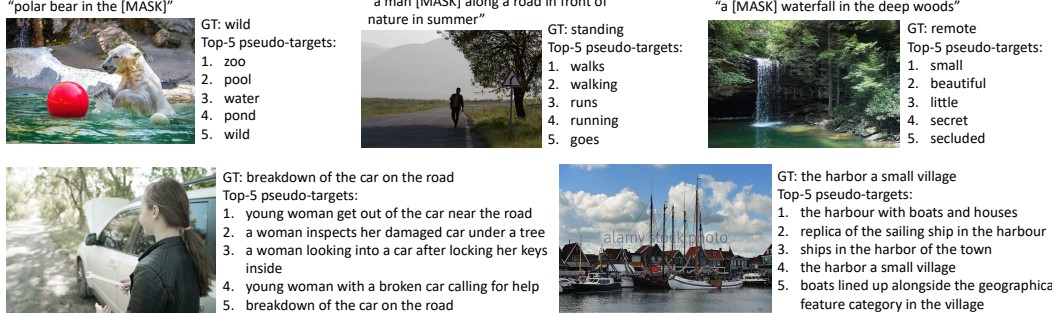

Figure 2: Examples of the pseudo-targets for MLM (1st row) and ITC (2nd row). The pseudo-targets can capture visual concepts that are not described by the ground-truth text (*e.g.* "beautiful waterfall", "young woman").

**Masked Language Modeling** utilizes both the image and the contextual text to predict the masked words. We randomly mask out the input tokens with a probability of 15% and replace them with the special token [MASK][1]. Let $\hat{T}$ denote a masked text, and $\boldsymbol{p}^{\mathrm{msk}}(I, \hat{T})$ denote the model's predicted probability for a masked token. MLM minimizes a cross-entropy loss:

$$\mathcal{L}_{\mathrm{mlm}} = \mathbb{E}_{(I,\hat{T})\sim D}\mathrm{H}(\boldsymbol{y}^{\mathrm{msk}}, \boldsymbol{p}^{\mathrm{msk}}(I, \hat{T})) \qquad (3)$$

where $\boldsymbol{y}^{\mathrm{msk}}$ is a one-hot vocabulary distribution where the ground-truth token has a probability of 1.

**Image-Text Matching** predicts whether a pair of image and text is positive (matched) or negative (not matched). We use the multimodal encoder's output embedding of the [CLS] token as the joint representation of the image-text pair, and append a fully-connected (FC) layer followed by softmax to predict a two-class probability $p^{\mathrm{itm}}$. The ITM loss is:

$$\mathcal{L}_{\mathrm{itm}} = \mathbb{E}_{(I,T)\sim D}\mathrm{H}(\boldsymbol{y}^{\mathrm{itm}}, \boldsymbol{p}^{\mathrm{itm}}(I, T)) \qquad (4)$$

where $\boldsymbol{y}^{\mathrm{itm}}$ is a 2-dimensional one-hot vector representing the ground-truth label.

We propose a strategy to sample hard negatives for the ITM task with zero computational overhead. A negative image-text pair is hard if they share similar semantics but differ in fine-grained details. We use the contrastive similarity from Equation 1 to find in-batch hard negatives. For each image in a mini-batch, we sample one negative text from the same batch following the contrastive similarity distribution, where texts that are more similar to the image have a higher chance to be sampled. Likewise, we also sample one hard negative image for each text.

The full pre-training objective of ALBEF is:

$$\mathcal{L} = \mathcal{L}_{\mathrm{itc}} + \mathcal{L}_{\mathrm{mlm}} + \mathcal{L}_{\mathrm{itm}} \qquad (5)$$

### 3.3 Momentum Distillation

The image-text pairs used for pre-training are mostly collected from the web and they tend to be noisy. Positive pairs are usually weakly-correlated: the text may contain words that are unrelated to the image, or the image may contain entities that are not described in the text. For ITC learning, negative texts for an image may also match the image's content. For MLM, there may exist other words different from the annotation that describes the image equally well (or better). However, the one-hot labels for ITC and MLM penalize all negative predictions regardless of their correctness.

To address this, we propose to learn from pseudo-targets generated by the momentum model. The momentum model is a continuously-evolving teacher which consists of exponential-moving-average versions of the unimodal and multimodal encoders. During training, we train the base model such that its predictions match the ones from the momentum model. Specifically, for ITC, we first compute the image-text similarity using features from the momentum unimodal encoders as $s'(I, T) = g'_v(\boldsymbol{v}'_{\mathrm{cls}})^{\top} g'_w(\boldsymbol{w}'_{\mathrm{cls}})$ and $s'(T, I) = g'_w(\boldsymbol{w}'_{\mathrm{cls}})^{\top} g'_v(\boldsymbol{v}'_{\mathrm{cls}})$. Then we compute soft pseudo-targets $\boldsymbol{q}^{\mathrm{i2t}}$ and $\boldsymbol{q}^{\mathrm{t2i}}$ by replacing $s$ with $s'$ in Equation 1. The ITC$_{\mathrm{MoD}}$ loss is defined as:

$$\mathcal{L}_{\mathrm{itc}}^{\mathrm{mod}} = (1 - \alpha)\mathcal{L}_{\mathrm{itc}} + \frac{\alpha}{2}\mathbb{E}_{(I,T)\sim D}\big[\mathrm{KL}(\boldsymbol{q}^{\mathrm{i2t}}(I) \parallel \boldsymbol{p}^{\mathrm{i2t}}(I)) + \mathrm{KL}(\boldsymbol{q}^{\mathrm{t2i}}(T) \parallel \boldsymbol{p}^{\mathrm{t2i}}(T))\big] \qquad (6)$$

---

[1]following BERT, the replacements are 10% random tokens, 10% unchanged, and 80% [MASK]

Similarly, for MLM, let $\boldsymbol{q}^{\mathrm{msk}}(I, \hat{T})$ denote the momentum model's prediction probability for the masked token, the $\mathrm{MLM_{MoD}}$ loss is:

$$\mathcal{L}_{\mathrm{mlm}}^{\mathrm{mod}} = (1 - \alpha)\mathcal{L}_{\mathrm{mlm}} + \alpha\mathbb{E}_{(I,\hat{T})\sim D}\mathrm{KL}(\boldsymbol{q}^{\mathrm{msk}}(I, \hat{T}) \parallel \boldsymbol{p}^{\mathrm{msk}}(I, \hat{T})) \tag{7}$$

In Figure 2, we show examples of the top-5 candidates from the pseudo-targets, which effectively capture relevant words/texts for an image. More examples can be found in Appendix.

We also apply MoD to the downstream tasks. The final loss for each task is a weighted combination of the original task's loss and the KL-divergence between the model's prediction and the pseudo-targets. For simplicity, we set the weight $\alpha = 0.4$ for all pre-training and downstream tasks [2].

### 3.4 Pre-training Datasets

Following UNITER [2], we construct our pre-training data using two web datasets (Conceptual Captions [4], SBU Captions [5]) and two in-domain datasets (COCO [41] and Visual Genome [42]). The total number of unique images is 4.0M, and the number of image-text pairs is 5.1M. To show that our method is scalable with larger-scale web data, we also include the much noisier Conceptual 12M dataset [43], increasing the total number of images to 14.1M [3]. Details are in Appendix.

### 3.5 Implementation Details

Our model consists of a $\mathrm{BERT_{base}}$ with 123.7M parameters and a ViT-B/16 with 85.8M parameters. We pre-train the model for 30 epochs using a batch size of 512 on 8 NVIDIA A100 GPUs. We use the AdamW [44] optimizer with a weight decay of 0.02. The learning rate is warmed-up to $1e^{-4}$ in the first 1000 iterations, and decayed to $1e^{-5}$ following a cosine schedule. During pre-training, we take random image crops of resolution $256 \times 256$ as input, and also apply RandAugment [4] [45]. During fine-tuning, we increase the image resolution to $384 \times 384$ and interpolate the positional encoding of image patches following [38]. The momentum parameter for updating the momentum model is set as 0.995, and the size of the queue used for image-text contrastive learning is set as 65,536. We linearly ramp-up the distillation weight $\alpha$ from 0 to 0.4 within the 1st epoch.

## 4 A Mutual Information Maximization Perspective

In this section, we provide an alternative perspective of ALBEF and show that it maximizes a lower bound on the mutual information (MI) between different "views" of an image-text pair. ITC, MLM, and MoD can be interpreted as different ways to generate the views.

Formally, we define two random variables $a$ and $b$ as two different views of a data point. In self-supervised learning [24, 25, 46], $a$ and $b$ are two augmentations of the same image. In vision-language representation learning, we consider $a$ and $b$ as different variations of an image-text pair that capture its semantic meaning. We aim to learn representations invariant to the change of view. This can be achieved by maximizing the MI between $a$ and $b$. In practice, we maximize a lower bound on $\mathrm{MI}(a, b)$ by minimizing the InfoNCE loss [47] defined as:

$$\mathcal{L}_{\mathrm{NCE}} = -\mathbb{E}_{p(a,b)}\left[\log\frac{\exp(s(a, b))}{\sum_{\hat{b}\in\hat{B}}\exp(s(a, \hat{b}))}\right] \tag{8}$$

where $s(a, b)$ is a scoring function (*e.g.*, a dot product between two representations), and $\hat{B}$ contains the positive sample $b$ and $|\hat{B}| - 1$ negative samples drawn from a proposal distribution.

Our ITC loss with one-hot labels (Equation 2) can be re-written as:

$$\mathcal{L}_{\mathrm{itc}} = -\frac{1}{2}\mathbb{E}_{p(I,T)}\Big[\log\frac{\exp(s(I, T)/\tau)}{\sum_{m=1}^{M}\exp(s(I, T_m)/\tau)} + \log\frac{\exp(s(T, I)/\tau)}{\sum_{m=1}^{M}\exp(s(T, I_m)/\tau)}\Big] \tag{9}$$

Minimizing $\mathcal{L}_{\mathrm{itc}}$ can be seen as maximizing a symmetric version of InfoNCE. Hence, ITC considers the two individual modalities (*i.e.*, $I$ and $T$) as the two views of an image-text pair, and trains the unimodal encoders to maximize the MI between the image and text views for the positive pairs.

---

[2] our experiments show that $\alpha = 0.3, 0.4, 0.5$ yield similar performance, with $\alpha = 0.4$ slightly better

[3] some urls provided by the web datasets have become invalid

[4] we remove color changes from RandAugment because the text often contains color information

As shown in [48], we can also interpret MLM as maximizing the MI between a masked word token and its masked context (*i.e.* image + masked text). Specifically, we can re-write the MLM loss with one-hot labels (Equation 3) as

$$\mathcal{L}_{\mathrm{mlm}} = -\mathbb{E}_{p(I,\hat{T})}\Big[\log\frac{\exp(\psi(y^{\mathrm{msk}})^{\top}f(I,\hat{T}))}{\sum_{y\in\mathcal{V}}\exp(\psi(y)^{\top}f(I,\hat{T}))}\Big] \tag{10}$$

where $\psi(y) : \mathcal{V} \to \mathbb{R}^d$ is a lookup function in the multimodal encoder's output layer that maps a word token $y$ into a vector and $\mathcal{V}$ is the full vocabulary set, and $f(I,\hat{T})$ is a function that returns the final hidden state of the multimodal encoder corresponding to the masked context. Hence, MLM considers the two views of an image-text pair to be: (1) a randomly selected word token, and (2) the image + the contextual text with that word masked.

Both ITC and MLM generate views by taking partial information from an image-text pair, through either modality separation or word masking. Our momentum distillation can be considered as generating alternative views from the entire proposal distribution. Take ITC$_{\mathrm{MoD}}$ in Equation 6 as an example, minimizing $\mathrm{KL}(\boldsymbol{p}^{\mathrm{i2t}}(I), \boldsymbol{q}^{\mathrm{i2t}}(I))$ is equivalent to minimizing the following objective:

$$-\sum_m q_m^{\mathrm{i2t}}(I)\log p_m^{\mathrm{i2t}}(I) = -\sum_m\frac{\exp(s'(I,T_m)/\tau)}{\sum_{m=1}^{M}\exp(s'(I,T_m)/\tau)}\log\frac{\exp(s(I,T_m)/\tau)}{\sum_{m=1}^{M}\exp(s(I,T_m)/\tau)} \tag{11}$$

It maximizes $\mathrm{MI}(I,T_m)$ for texts that share similar semantic meaning with the image $I$ because those texts would have larger $q_m^{\mathrm{i2t}}(I)$. Similarly, ITC$_{\mathrm{MoD}}$ also maximizes $\mathrm{MI}(I_m,T)$ for images that are similar to $T$. We can follow the same method to show that MLM$_{\mathrm{MoD}}$ generates alternative views $y' \in \mathcal{V}$ for the masked word $y^{\mathrm{msk}}$, and maximizes the MI between $y'$ and $(I,\hat{T})$. Therefore, our momentum distillation can be considered as performing data augmentation to the original views. The momentum model generates a diverse set of views that are absent in the original image-text pairs, and encourages the base model to learn representations that capture view-invariant semantic information.

## 5 Downstream V+L Tasks

We adapt the pre-trained model to five downstream V+L tasks. We introduce each task and our fine-tuning strategy below. Details of the datasets and fine-tuning hyperparameters are in Appendix.

**Image-Text Retrieval** contains two subtasks: image-to-text retrieval (TR) and text-to-image retrieval (IR). We evaluate ALBEF on the Flickr30K [49] and COCO benchmarks, and fine-tune the pre-trained model using the training samples from each dataset. For zero-shot retrieval on Flickr30K, we evaluate with the model fine-tuned on COCO. During fine-tuning, we jointly optimize the ITC loss (Equation 2) and the ITM loss (Equation 4). ITC learns an image-text scoring function based on similarity of unimodal features, whereas ITM models the fine-grained interaction between image and text to predict a matching score. Since the downstream datasets contain multiple texts for each image, we change the ground-truth label of ITC to consider multiple positives in the queue, where each positive has a ground-truth probability of $1/\#\mathrm{positives}$. During inference, we first compute the feature similarity score $s_{\mathrm{itc}}$ for all image-text pairs. Then we take the top-$k$ candidates and calculate their ITM score $s_{\mathrm{itm}}$ for ranking. Because $k$ can be set to be very small, our inference speed is much faster than methods that require computing the ITM score for all image-text pairs [2, 3, 8].

**Visual Entailment** (SNLI-VE[5] [51]) is a fine-grained visual reasoning task to predict whether the relationship between an image and a text is entailment, neutral, or contradictory. We follow UNITER [2] and consider VE as a three-way classification problem, and predict the class probabilities using a multi-layer perceptron (MLP) on the multimodal encoder's representation of the [CLS] token.

**Visual Question Answering** (VQA [52]) requires the model to predict an answer given an image and a question. Different from existing methods that formulate VQA as a multi-answer classification problem [53, 2], we consider VQA as an answer generation problem, similar to [54]. Specifically, we use a 6-layer transformer decoder to generate the answer. As shown in Figure 3a, the auto-regressive answer decoder receives the multimodal embeddings through cross attention, and a start-of-sequence token ([CLS]) is used as the decoder's initial input token. Likewise, an end-of-sequence token ([SEP]) is appended to the end of decoder outputs which indicates the completion of generation.

---

[5]results on SNLI-VE should be interpreted with caution because its test data has been reported to be noisy [50]

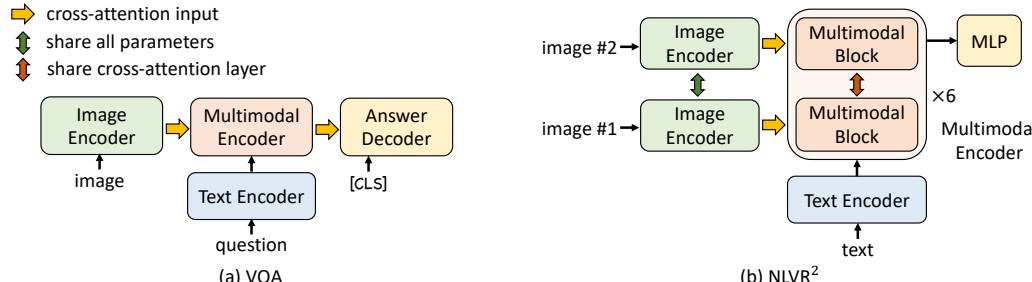

Figure 3: The model architecture for VQA and NLVR$^2$. For VQA, we append an auto-regressive decoder to generate the answer given the image-question embeddings. For NLVR$^2$, we replicate the transformer block within each layer of multimodal encoder to enable reasoning over two images.

The answer decoder is initialized using the pre-trained weights from the multimodal encoder, and finetuned with a conditional language-modeling loss. For a fair comparison with existing methods, we constrain the decoder to only generate from the 3,128 candidate answers [55] during inference.

**Natural Language for Visual Reasoning** (NLVR$^2$ [19]) requires the model to predict whether a text describes a pair of images. We extend our multimodal encoder to enable reasoning over two images. As shown in Figure 3b, each layer of the multimodal encoder is replicated to have two consecutive transformer blocks, where each block contains a self-attention layer, a cross-attention layer, and a feed-forward layer (see Figure 1). The two blocks within each layer are initialized using the same pre-trained weights, and the two cross-attention layers share the same linear projection weights for the keys and values. During training, the two blocks receive two sets of image embeddings for the image pair. We append a MLP classifier on the multimodal encoder's [CLS] representation for prediction.

For NLVR$^2$, we perform an additional pre-training step to prepare the new multimodal encoder for encoding an image-pair. We design a text-assignment (TA) task as follows: given a pair of images and a text, the model needs to assign the text to either the first image, the second image, or none of them. We consider it as a three-way classification problem, and use a FC layer on the [CLS] representation to predict the assignment. We pre-train with TA for only 1 epoch using the 4M images (Section 3.4).

**Visual Grounding** aims to localize the region in an image that corresponds to a specific textual description. We study the weakly-supervised setting, where no bounding box annotations are available. We perform experiments on the RefCOCO+ [56] dataset, and fine-tune the model using only image-text supervision following the same strategy as image-text retrieval. During inference, we extend Grad-CAM [9] to acquire heatmaps, and use them to rank the detected proposals provided by [53].

## 6 Experiments

### 6.1 Evaluation on the Proposed Methods

First, we evaluate the effectiveness of the proposed methods (*i.e.* image-text contrastive learning, contrastive hard negative mining, and momentum distillation). Table 1 shows the performance of the downstream tasks with different variants of our method. Compared to the baseline pre-training tasks (MLM+ITM), adding ITC substantially improves the pre-trained model's performance across

| #Pre-train Images | Training tasks | TR (flickr test) | IR (flickr test) | SNLI-VE (test) | NLVR$^2$ (test-P) | VQA (test-dev) |
|---|---|---|---|---|---|---|
| 4M | MLM + ITM | 93.96 | 88.55 | 77.06 | 77.51 | 71.40 |
| | ITC + MLM + ITM | 96.55 | 91.69 | 79.15 | 79.88 | 73.29 |
| | ITC + MLM + ITM$_{hard}$ | 97.01 | 92.16 | 79.77 | 80.35 | 73.81 |
| | ITC$_{MoD}$ + MLM + ITM$_{hard}$ | 97.33 | 92.43 | 79.99 | 80.34 | 74.06 |
| | Full (ITC$_{MoD}$ + MLM$_{MoD}$ + ITM$_{hard}$) | 97.47 | 92.58 | 80.12 | 80.44 | 74.42 |
| | ALBEF (Full + MoD$_{Downstream}$) | 97.83 | 92.65 | 80.30 | 80.50 | 74.54 |
| 14M | ALBEF | 98.70 | 94.07 | 80.91 | 83.14 | 75.84 |

Table 1: Evaluation of the proposed methods on four downstream V+L tasks. For text-retrieval (TR) and image-retrieval (IR), we report the average of R@1, R@5 and R@10. ITC: image-text contrastive learning. MLM: masked language modeling. ITM$_{hard}$: image-text matching with contrastive hard negative mining. MoD: momentum distillation. MoD$_{Downstream}$: momentum distillation on downstream tasks.

| Method | # Pre-train Images | Flickr30K (1K test set) | | | | | | MSCOCO (5K test set) | | | | | |
|---|---|---|---|---|---|---|---|---|---|---|---|---|---|
| | | TR | | | IR | | | TR | | | IR | | |
| | | R@1 | R@5 | R@10 | R@1 | R@5 | R@10 | R@1 | R@5 | R@10 | R@1 | R@5 | R@10 |
| UNITER | 4M | 87.3 | 98.0 | 99.2 | 75.6 | 94.1 | 96.8 | 65.7 | 88.6 | 93.8 | 52.9 | 79.9 | 88.0 |
| VILLA | 4M | 87.9 | 97.5 | 98.8 | 76.3 | 94.2 | 96.8 | - | - | - | - | - | - |
| OSCAR | 4M | - | - | - | - | - | - | 70.0 | 91.1 | 95.5 | 54.0 | 80.8 | 88.5 |
| ALIGN | 1.2B | 95.3 | 99.8 | 100.0 | 84.9 | 97.4 | 98.6 | 77.0 | 93.5 | 96.9 | 59.9 | 83.3 | 89.8 |
| ALBEF | 4M | 94.3 | 99.4 | 99.8 | 82.8 | 96.7 | 98.4 | 73.1 | 91.4 | 96.0 | 56.8 | 81.5 | 89.2 |
| ALBEF | 14M | **95.9** | **99.8** | **100.0** | **85.6** | **97.5** | **98.9** | **77.6** | **94.3** | **97.2** | **60.7** | **84.3** | **90.5** |

Table 2: Fine-tuned image-text retrieval results on Flickr30K and COCO datasets.

| Method | # Pre-train Images | Flickr30K (1K test set) | | | | | |
|---|---|---|---|---|---|---|---|
| | | TR | | | IR | | |
| | | R@1 | R@5 | R@10 | R@1 | R@5 | R@10 |
| UNITER [2] | 4M | 83.6 | 95.7 | 97.7 | 68.7 | 89.2 | 93.9 |
| CLIP [6] | 400M | 88.0 | 98.7 | 99.4 | 68.7 | 90.6 | 95.2 |
| ALIGN [7] | 1.2B | 88.6 | 98.7 | 99.7 | 75.7 | 93.8 | 96.8 |
| ALBEF | 4M | 90.5 | 98.8 | 99.7 | 76.8 | 93.7 | 96.7 |
| ALBEF | 14M | **94.1** | **99.5** | 99.7 | **82.8** | **96.3** | **98.1** |

Table 3: Zero-shot image-text retrieval results on Flickr30K.

| Method | VQA | | NLVR$^2$ | | SNLI-VE | |
|---|---|---|---|---|---|---|
| | test-dev | test-std | dev | test-P | val | test |
| VisualBERT [13] | 70.80 | 71.00 | 67.40 | 67.00 | - | - |
| VL-BERT [10] | 71.16 | - | - | - | - | - |
| LXMERT [1] | 72.42 | 72.54 | 74.90 | 74.50 | - | - |
| 12-in-1 [12] | 73.15 | - | - | 78.87 | - | 76.95 |
| UNITER [2] | 72.70 | 72.91 | 77.18 | 77.85 | 78.59 | 78.28 |
| VL-BART/T5 [54] | - | 71.3 | - | 73.6 | - | - |
| ViLT [21] | 70.94 | - | 75.24 | 76.21 | - | - |
| OSCAR [3] | 73.16 | 73.44 | 78.07 | 78.36 | - | - |
| VILLA [8] | 73.59 | 73.67 | 78.39 | 79.30 | 79.47 | 79.03 |
| ALBEF (4M) | 74.54 | 74.70 | 80.24 | 80.50 | 80.14 | 80.30 |
| ALBEF (14M) | **75.84** | **76.04** | **82.55** | **83.14** | **80.80** | **80.91** |

Table 4: Comparison with state-of-the-art methods on downstream vision-language tasks.

all tasks. The proposed hard negative mining improves ITM by finding more informative training samples. Furthermore, adding momentum distillation improves learning for both ITC (row 4), MLM (row 5), and on all downstream tasks (row 6). In the last row, we show that ALBEF can effectively leverage more noisy web data to improve the pre-training performance.

## 6.2 Evaluation on Image-Text Retrieval

Table 2 and Table 3 report results on fine-tuned and zero-shot image-text retrieval, respectively. Our ALBEF achieves state-of-the-art performance, outperforming CLIP [6] and ALIGN [7] which are trained on orders of magnitude larger datasets. Given the considerable amount of improvement of ALBEF when the number of training images increases from 4M to 14M, we hypothesize that it has potential to further grow by training on larger-scale web image-text pairs.

## 6.3 Evaluation on VQA, NLVR, and VE

Table 4 reports the comparison with existing methods on other V+L understanding tasks. With 4M pre-training images, ALBEF already achieves state-of-the-art performance. With 14M pre-training images, ALBEF substantially outperforms existing methods, including methods that additionally use object tags [3] or adversarial data augmentation [8]. Compared to VILLA [8], ALBEF achieves absolute improvements of 2.37% on VQA test-std, 3.84% on NLVR$^2$ test-P, and 1.88% on SNLI-VE test. Because ALBEF is detector-free and requires lower resolution images, it also enjoys much faster inference speed compared to most existing methods (>10 times faster than VILLA on NLVR$^2$).

## 6.4 Weakly-supervised Visual Grounding

Table 5 shows the results on RefCOCO+, where ALBEF substantially outperforms existing methods [57, 58] (which use weaker text embeddings). The ALBEF$_{itc}$ variant computes Grad-CAM

| Method | Val | TestA | TestB |
|---|---|---|---|
| ARN [57] | 32.78 | 34.35 | 32.13 |
| CCL [58] | 34.29 | 36.91 | 33.56 |
| ALBEF$_{itc}$ | 51.58 | 60.09 | 40.19 |
| ALBEF$_{itm}$ | **58.46** | **65.89** | **46.25** |

Table 5: Weakly-supervised visual grounding on RefCOCO+ [56] dataset.

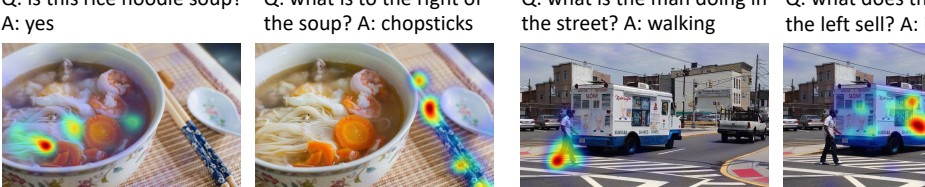

Figure 4: Grad-CAM visualization on the cross-attention maps in the 3rd layer of the multimodal encoder.

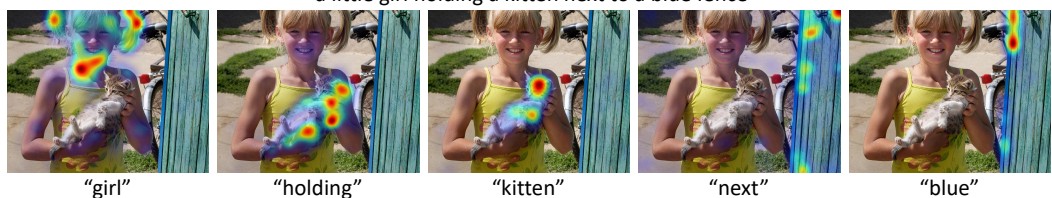

Figure 5: Grad-CAM visualizations on the cross-attention maps of the multimodal encoder for the VQA model.

"a little girl holding a kitten next to a blue fence"

Figure 6: Grad-CAM visualizations on the cross-attention maps corresponding to individual words.

visualizations on the self-attention maps in the last layer of the image encoder, where the gradients are acquired by maximizing the image-text similarity $s_{itc}$. The ALBEF$_{itm}$ variant computes Grad-CAM on the cross-attention maps in the 3rd layer of the multimodal encoder (which is a layer specialized in grounding), where the gradients are acquired by maximizing the image-text matching score $s_{itm}$. Figure 4 provides a few visualizations. More analysis is in Appendix.

We provide the Grad-CAM visualizations for VQA in Figure 5. As can be seen in Appendix, the Grad-CAM visualizations from ALBEF are highly correlated with where humans would look when making decisions. In Figure 6, we show per-word visualizations for COCO. Notice how our model not only grounds objects, but also their attributes and relationships.

## 6.5 Ablation Study

Table 6 studies the effect of various design choices on image-text retrieval. Since we use $s_{itc}$ to filter top-$k$ candidates during inference, we vary $k$ and report its effect. In general, the ranking result acquired by $s_{itm}$ is not sensitive to changes in $k$. We also validate the effect of hard negative mining in the last column.

| Flickr30K | | w/ hard negs | | | w/o hard negs |
|---|---|---|---|---|---|
| | $s_{itc}$ | $k = 16$ | $k = 128$ | $k = 256$ | $k = 128$ |
| TR | 97.30 | 98.60 | 98.57 | 98.57 | 98.22 (−0.35) |
| IR | 90.95 | 93.64 | 93.99 | 93.95 | 93.68 (−0.31) |

Table 6: Ablation study on fine-tuned image-text retrieval. The average recall on the test set is reported. We use $s_{itc}$ to filter top-$k$ candidates and calculate their $s_{itm}$ score for ranking.

Table 7 studies the effect of text-assignment (TA) pre-training and parameter sharing on NLVR$^2$. We examine three strategies: (1) the two mutimodal blocks share all parameters, (2) only the cross-attention (CA) layers are shared, (3) no sharing. Without TA, sharing the entire block has better

| NLVR$^2$ | w/ TA | | | w/o TA | | |
|---|---|---|---|---|---|---|
| | share all | share CA | no share | share all | share CA | no share |
| dev | 82.13 | 82.55 | 81.93 | 80.52 | 80.28 | 77.84 |
| test-P | 82.36 | 83.14 | 82.85 | 81.29 | 80.45 | 77.58 |

Table 7: Ablation study on NLVR$^2$.

performance. With TA to pre-train the model for image-pair, sharing CA leads to the best performance.

## 7 Conclusion and Social Impacts

This paper proposes ALBEF, a new framework for vision-language representation learning. ALBEF first aligns the unimodal image representation and text representation before fusing them with a multimodal encoder. We theoretically and experimentally verify the effectiveness of the proposed

image-text contrastive learning and momentum distillation. Compared to existing methods, ALBEF offers better performance and faster inference speed on multiple downstream V+L tasks.

While our paper shows promising results on vision-language representation learning, additional analysis on the data and the model is necessary before deploying it in practice, because web data may contain unintended private information, unsuitable images, or harmful texts, and only optimizing accuracy may have unwanted social implications.

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
