# A    Downstream Task Details

Here we describe the implementation details for fine-tuning the pre-trained model. For all downstream tasks, we use the same RandAugment, AdamW optimizer, cosine learning rate decay, weight decay, and distillation weight as during pre-training. All downstream tasks receive input images of resolution $384 \times 384$. During inference, we resize the images without any cropping.

**Image-Text Retrieval.** We consider two datasets for this task: COCO and Flickr30K. We adopt the widely used Karpathy split [59] for both datasets. COCO contains 113/5k/5k for train/validation/test. Flickr30K contains 29k/1k/1k images for train/validation/test. We fine-tune for 10 epochs. The batch size is 256 and the initial learning rate is $1e^{-5}$.

**Visual Entailment.** We evaluate on the SNLI-VE dataset [51], which is constructed using the Stanford Natural Language Inference (SNLI) [60] and Flickr30K datasets. We follow the original dataset split with 29.8k images for training, 1k for evaluation, and 1k for test. We fine-tune the pre-trained model for 5 epochs with a batch size of 256 and an initial learning rate of $2e^{-5}$.

**VQA.** We conduct experiment on the VQA2.0 dataset [52], which is constructed using images from COCO. It contains 83k images for training, 41k for validation, and 81k for test. We report performance on the test-dev and test-std splits. Following most existing works [1, 2, 61], we use both training and validation sets for training, and include additional question-answer pairs from Visual Genome. Because many questions in the VQA dataset contains multiple answers, we weight the loss for each answer by its percentage of occurrence among all answers. We fine-tune the model for 8 epochs, using a batch size of 256 and an initial learning rate of $2e^{-5}$.

**NLVR$^2$.** We conduct experiments following the original train/val/test split in [19]. We fine-tune the model for 10 epochs, using a batch size of 128 and an initial learning rate of $2e^{-5}$. Because NLVR receives two input images, we perform an additional step of pre-training with text-assignment (TA) to prepare the model for reasoning over two images. The TA pre-training uses images of size $256 \times 256$. We pre-train for 1 epoch on the 4M dataset, using a batch size of 256 and a learning rate of $2e^{-5}$.

**Visual Grounding.** We conduct experiments on the RefCOCO+ dataset [56], which is collected using a two-player ReferitGame [62]. It contains 141,564 expressions for 19,992 images from COCO training set. Strictly speaking, our model is not allowed to see the val/test images of RefCOCO+, but it has been exposed to those images during pre-training. We hypothesize that this has little effect because these images only occupy a very small portion of the entire 14M pre-training images, and leave it as future work to decontaminate the data. During weakly-supervised fine-tuning, we follow the same strategy as image-text retrieval except that we do not perform random cropping, and train the model for 5 epochs. During inference, we use either $s_{\text{itc}}$ or $s_{\text{itm}}$ to compute the importance score for each $16 \times 16$ image patch. For ITC, we compute Grad-CAM visualizations on the self-attention maps *w.r.t* the `[CLS]` token in the last layer of the visual encoder, and average the heatmaps across all attention heads. For ITM, we compute Grad-CAM on the cross-attention maps in the 3rd layer of the multimodal encoder, and average them scores across all attention heads and all input text tokens. Quantitative comparison between ITC and ITM is shown in Table 5. Figure 7 shows the qualitative comparison. Since the multimodal encoder can better model image-text interactions, it produces better heatmaps that capture finer-grained details. In Figure 8, we report the grounding accuracy for each cross-attention layer and each individual attention head within the best-performing layer.

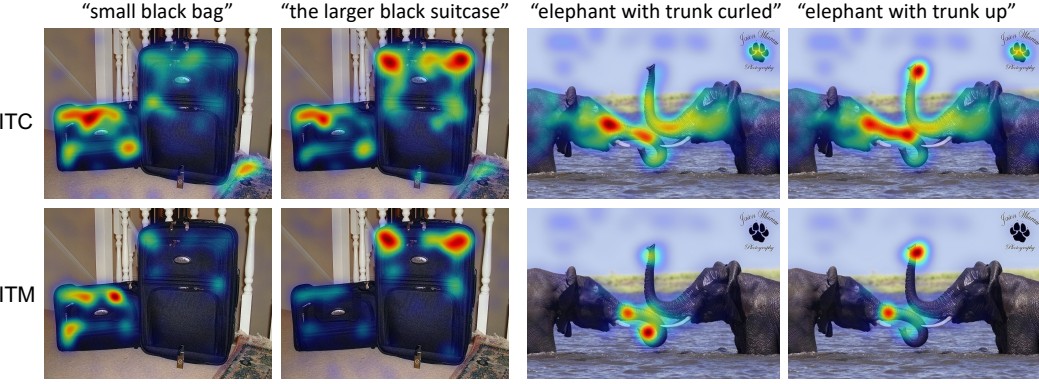

Figure 7: Grad-CAMs from the multimodal encoder capture finer-grained details such as "larger" and "curled".

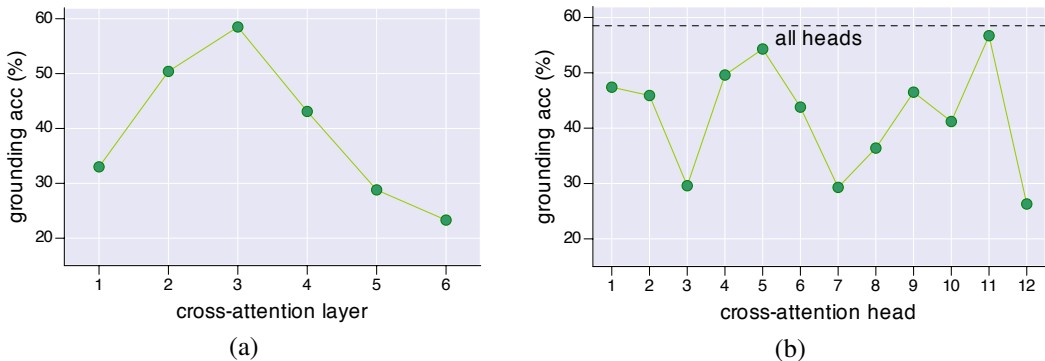

(a)              (b)

Figure 8: Grounding accuracy on the validation set of RefCOCO+. (a) varying cross-attention layers where each layer uses all heads. (b) varying cross-attention heads in the best-performing (3rd) layer.

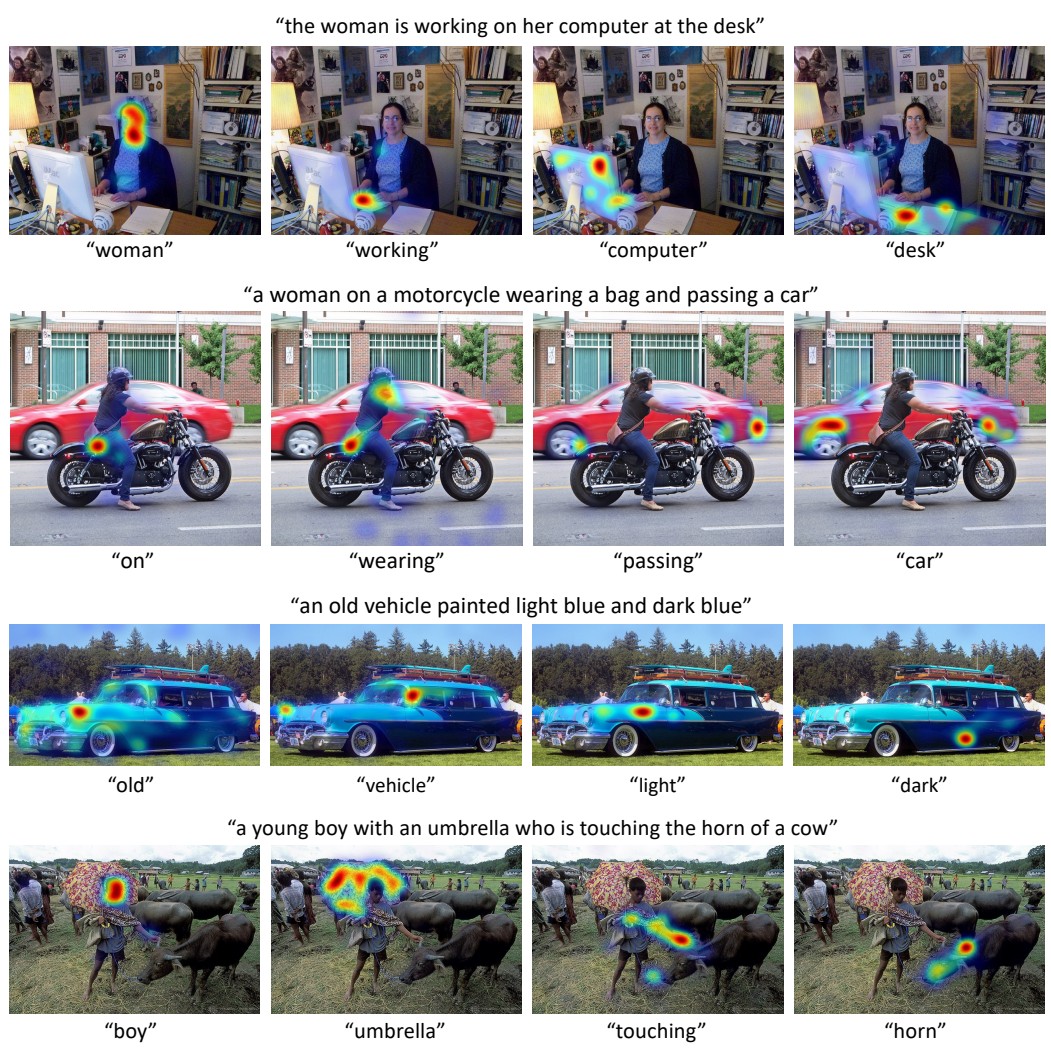

Figure 9: Grad-CAM visualization on the cross-attention maps corresponding to individual words.

## B  Additional Per-word Visualizations

In Figure 9, we show more visualizations of per-word Grad-CAM to demonstrate the ability of our model to perform visual grounding of objects, actions, attributes, and relationships.

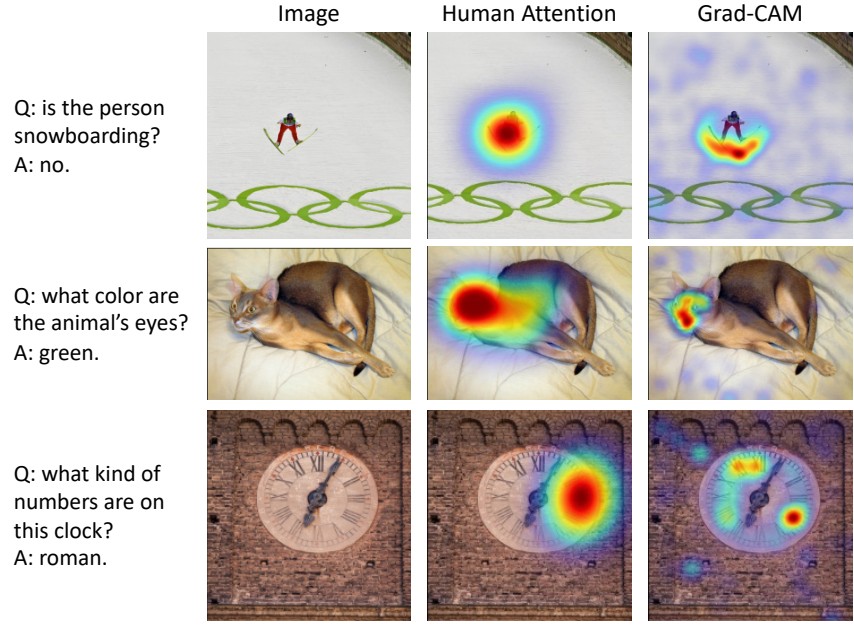

Figure 10: Qualitative comparison between human attention and ALBEF's Grad-CAM for VQA.

## C  Comparison with Human Attention

Das *et al.* [63] collected human attention maps for a subset of the VQA dataset [20]. Given a question and a blurred version of the image, humans on Amazon Mechanical Turk were asked to interactively deblur image regions until they could confidently answer the question. In this work we compare human attention maps to Grad-CAM visualizations for the ALBEF VQA model computed at the 3rd multi-modal cross-attention layer on 1374 validation question-image pairs using the rank correlation evaluation protocol as in [63]. We find Grad-CAM and human attention maps computed for the ground-truth answer to have a high correlation of 0.205. This shows that despite not being trained on grounded image-text pairs, ALBEF looks at appropriate regions when making decisions. Qualitative examples showing the comparison with human attention maps can be found in Figure 10.

## D  Additional Examples of Pseudo-targets

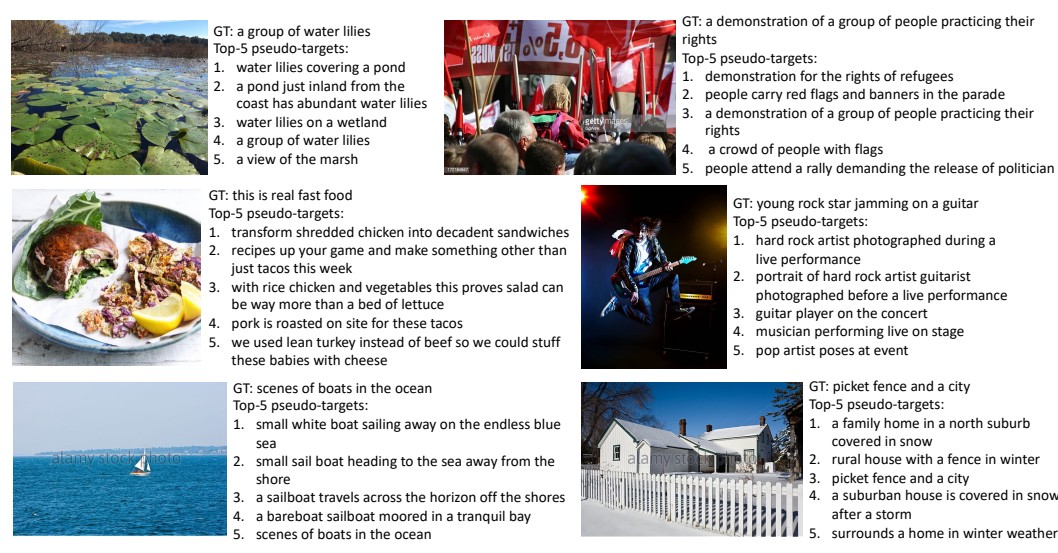

Figure 11: Examples of the top-5 most similar texts selected by the momentum model for ITC.

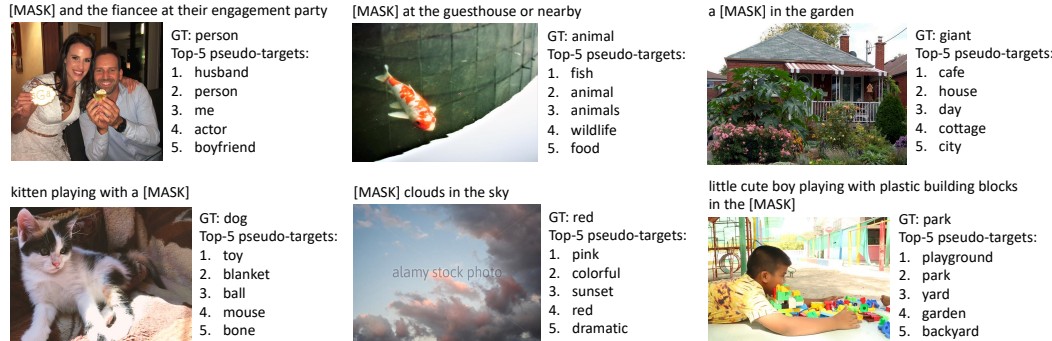

Figure 12: Examples of the top-5 words generated by the momentum model for MLM.

# E  Pre-training Dataset Details

Table 8 shows the statistics of the image and text of the pre-training datasets.

|  | COCO (Karpathy-train) | VG | CC | SBU | CC12M |
|---|---|---|---|---|---|
| # image | 113K | 100K | 2.95M | 860K | 10.06M |
| # text | 567K | 769K | 2.95M | 860K | 10.06M |

Table 8: Statistics of the pre-training datasets.