# OpenReview forum: "Align before Fuse: Vision and Language Representation Learning with Momentum Distillation"
_NeurIPS.cc/2021/Conference — NeurIPS 2021 Spotlight_

### Official Review · Reviewer_16yc · 2021-07-14

**Rating:** 7
**Confidence:** 4

**Summary:**

In this paper, the authors study the vision-transformer-like vision-language-pretraining (VLP) methods. The raw image and text are first encoded by a vision transformer and a text transformer respectively, and are then sent into a multimodal transformer for fusion. There are two major technical contributions, i.e., 1) the proposed ITC loss and 2) the moving average “teacher” to generate the pseudo targets.


The proposed method (ALBEF) outperforms previous E2E VLP methods, and achieves comparable performance to the VLP methods that take object region features.


**Limitations And Societal Impact:**

The limitations and societal impacts are discussed in the conclusion section.

**Main Review:**

Strengths:
1. This is a solid work that improves the SOTA of the VLP with raw image input.

2. The proposed image-text contrastive learning (ITC) and moving average teacher are intuitive and effectively improve the model performance. To my knowledge, the idea of aligning the visual and text modalities before sending them into the multimodal fusion as in ITC is new.

3. The analyses in Section 4 and the Grad-CAM visualization provide good analyses of the proposed approach.

Weaknesses:
1. The first two rows in Table 1 show that the proposed ITC loss significantly improves the performance. I wish to hear discussions and comments on where the ITC could be useful. Is it specified to the proposed vision-transformer-related VLP models, or it could be extended to the Vilt-like structure [21] with linear projection only, or even to the conventional VLP methods [13,2,3] with detector features.

2. The ablation of the model architecture might be necessary to understand the source of improvements, i.e., from the proposed losses and distillation, or the change in model architecture and patch size. Specifically, it might be necessary to ablate 1) different model sizes, e.g., small, base, large; 2) different patch sizes and numbers, e.g., ViT/32, /16, and input image size; 3) the influence of model initialization in Sec 3.1.

3. A minor concern is the comparison to SOTA in Table 4. The compared methods are the BASE version of the SOTA [3,8]. The LARGE version of [3,8] yields similar performance to the reported numbers.

4. A comment: The contribution of the momentum distillation and the analyses in Section 4 limit at applying the related techniques (moving average teacher in semi-supervised learning [33], the mutual information maximization perspective in NLP [A]) onto the VLP methods. Nonetheless, the proposed techniques and analyses do help the VLP study.

[A] A mutual information maximization perspective of language representation learning


**Time Spent Reviewing:**

4

---

> ### Author Response · Authors · 2021-08-10
> **Response**
>
> We appreciate the reviewer for the positive and insightful feedback. Our response to the reviewer’s questions is as follows.
>
> 1. *I wish to hear discussions and comments on where the ITC could be useful. Is it specified to the proposed vision-transformer-related VLP models, or it could be extended to the Vilt-like structure [21] with linear projection only, or even to the conventional VLP methods [13,2,3] with detector features.*
>
> The ITC loss helps VLP models to learn better unimodal representations that are (1) aligned with each other, and (2) better capture the semantic meaning of images and texts . Our proposed model architecture maximizes the advantage of ITC by feeding the aligned unimodal representations to a multimodal encoder (i.e., align before fuse). As the reviewer suggested, ITC could also be applied to other methods which use a shared multimodal transformer encoder for both image and text (e.g. ViLT [21], UNITER [2], etc.). In order to apply ITC, one could extract unimodal representations using the multimodal encoder (by separately feeding in each modality), and align the image and text representations. However, using the multimodal encoder to encode unimodal data may hurt its ability to learn image-text interactions.
>
> 2. *The ablation of the model architecture might be necessary to understand the source of improvements, i.e., from the proposed losses and distillation, or the change in model architecture and patch size. Specifically, it might be necessary to ablate 1) different model sizes, e.g., small, base, large; 2) different patch sizes and numbers, e.g., ViT/32, /16, and input image size; 3) the influence of model initialization in Sec 3.1.*
>
> 3. *A minor concern is the comparison to SOTA in Table 4. The compared methods are the BASE version of the SOTA [3,8]. The LARGE version of [3,8] yields similar performance to the reported numbers.*
>
> First, we would like to highlight that our proposed loss and distillation give clear and significant improvements as shown in Table 1.  Second, different from existing SoTA [3,8], our method does not require an object detector, therefore our ViT+BERT architecture is substantially more efficient. Third, our method uses BERT$_\mathrm{base}$, hence it is fair to compare our method with the BASE version of the SoTA methods which also use BERT$_\mathrm{base}$. Fourth, we have conducted experiments on a subset of pre-training data using higher image resolution, which gives better results. However, we chose to use a smaller resolution to improve efficiency.
>
> We appreciate the reviewer’s valuable suggestion on the ablation. We plan to conduct pre-training with better image and text backbones, which we expect to further improve performance. However, since pre-training and fine-tuning BERT$_\mathrm{large}$ is computationally expensive, we believe that BERT$_\mathrm{base}$ is a more practical choice for the community.

---

> > ### Comment · Reviewer_16yc · 2021-08-26
> > **Post Rebuttal**
> >
> > Thank you for your response. After reading other reviewer's comments and the author's responses, I would like to remain my initial positive rating of accept.

---

### Official Review · Reviewer_z2Jk · 2021-07-15

**Rating:** 7
**Confidence:** 3

**Summary:**

The paper proposes to learn joint vision and language representations by addressing several limitations in the existing methods. Firstly, they learn separate image and text representations using unimodal encoders without the need for bounding box annotations. Second, they fuse these two representations using a multimodal encoder based on cross modal attention and, third, they address the noise in large scale web based image-text datasets using a Momentum Distillation approach that generates pseudo-targets as additional supervision. They demonstrate the effectiveness of their approach on various downstream V+L tasks and gain substantial improvement over SOTA methods. Their main contributions are: removing the need for pre-trained object detector and high resolution images and combining it with the contrastive loss function for learning effective multi-modal representations, image-text contrastive learning loss (ITC). They also generate pseudo-targets for the ITC loss and the masked language modeling (MLM) loss using the momentum encoder model to address the weak correlations in the noisy image-text web data. This achieves high performance on both reasoning and retrieval tasks unlike other methods. The paper  is very well written and easy to follow and understand.


**Limitations And Societal Impact:**

The authors show that their proposed method gains improvement with significantly fewer images (14M) as compared to the state-of-the-art ALIGN model that uses 1.2B images to train. They also discuss the negative impact of the work due to leakage of private information, social implications, etc if deployed in real world settings. These things have to be taken care of when using such models in practice.


**Main Review:**

The ALBEF model proposed in the paper learns joint vision and language representations using three pretext tasks/objective functions - image-text contrastive learning (ITC), masked language modeling (MLM) and image-text matching (ITM). These pre-trained embeddings are then fine-tuned on several V+L downstream tasks to show their effectiveness. The method is significantly novel and addresses the limitations of existing work. It shows consistent improvements over SOTA methods by using relatively fewer data samples for pre-training.

1. In line 101, is w_cls the [CLS] token for the text inputs? For ITC loss, the similarity is computed based on these CLS embeddings, what would happen if instead it is computed based on the mean of the region/word-level embeddings?

2. Can the loss function in Eq 1., be added at the multimodal encoder stage as well?

3. It is not very clear how the momentum distillation learns to generate pseudo-targets. What is g’_v and g-_w in lines 148? Are these the average of the visual/textual features in the queue? How do they correspond to the image-text query to generate meaningful targets ?

4. Are the outputs from the ITC loss in Eq 6, softer after adding the KL divergence term ? If so, is this an advantage over existing methods for MLM and ITC? Would doing label smoothing or adding softmax temperature yield similar advantages?

5. In line 154-155, the authors say that they use  distillation loss for the downstream tasks, how does that work? Are the weights of this model fixed or fine-tuned? Or is it learned separately with the downstream task’s model?

6. In VQA downstream task, lines 234, are both the answer decoder and multimodal encoder initialized with the weights of multimodal encoder?



**Time Spent Reviewing:**

3.5

---

> ### Author Response · Authors · 2021-08-10
> **Response**
>
> We appreciate the reviewer for the positive and insightful feedback. Our response to the reviewer’s questions is as follows.
>
> 1. *In line 101, is w_cls the [CLS] token for the text inputs? For ITC loss, the similarity is computed based on these CLS embeddings, what would happen if instead it is computed based on the mean of the region/word-level embeddings?*
>
> $w_{cls}$ is the output text embedding for the [CLS] token. We performed preliminary experiments using the mean embeddings instead of CLS embeddings for the ITC loss, which leads to less improvement. The reason could be that the CLS embeddings enjoy more flexibility to capture the global representations of images/texts, which allow the model more capacity to jointly optimize multiple pre-training objectives.
>
> 2. *Can the loss function in Eq 1 be added at the multimodal encoder stage as well?*
>
> Theoretically, the image-text contrastive loss in Equation 1 can be applied to the multimodal encoder, where the similarity score $s(I,T)$ can be computed by applying a linear layer on top of the multimodal representation. However, as ITC requires a large number of negative pairs, it is infeasible to forward pass all the negative pairs through the multimodal encoder due to GPU memory limitation. Therefore, we use the binary image-text matching (ITM) loss instead, which requires a smaller number of negative pairs.
>
> 3. *It is not very clear how the momentum distillation learns to generate pseudo-targets. What is g’_v and g’_w in lines 148? Are these the average of the visual/textual features in the queue? How do they correspond to the image-text query to generate meaningful targets?*
>
> We generate pseudo-targets using the momentum encoders, whose parameters are the exponential moving average (EMA) of the encoders. Therefore, the momentum encoders are not trained with gradient descent. Instead, they are updated each time we update the encoders. Essentially, the momentum encoders can be considered as temporal ensembles of the encoders being trained.
>
> $v’_{cls}$ and $w’_{cls}$ represent the output [CLS] embeddings from the image and text momentum encoders, respectively. $g’_{v}$ and $g’_{w}$ are linear transformations which map $v’_{cls}$ and $w’_{cls}$ into normalized low-dimensional embeddings. These low-dimensional momentum embeddings are stored in two queues. For the standard ITC, the similarity of an image/text to a text/image is defined as the dot product between the **encoder’s embedding** and the **momentum encoder’s embedding**, as described in line 113.
>
> In order to generate pseudo-targets for ITC, we compute similarity between the current sample’s **momentum embedding** w.r.t the **momentum embeddings** in the queue. In other words, we find similar samples in the queue based on the momentum encoders only.
>
> 4. *Are the outputs from the ITC loss in Eq 6, softer after adding the KL divergence term? If so, is this an advantage over existing methods for MLM and ITC? Would doing label smoothing or adding softmax temperature yield similar advantages?*
>
> The advantage of adding the KL divergence term (i.e., momentum distillation) is to encourage the model to learn from other reasonable supervision that is not available in the noisy web annotation. As an effect of it, the model’s outputs do become softer. Different from the soft labels produced by the EMA model, standard label smoothing assigns uniform probabilities to all negatives, which neglects the difference of similarity across samples. We performed experiments which show that label smoothing does not yield similar advantages as our momentum distillation. In terms of softmax temperature, our ITC loss already contains a learnable temperature parameter $\tau$.
>
> 5. *In line 154-155, the authors say that they use distillation loss for the downstream tasks, how does that work? Are the weights of this model fixed or fine-tuned? Or is it learned separately with the downstream task’s model?*
>
> On the downstream tasks, we also maintain an EMA model as the teacher to generate pseudo-targets for the student. The weights of the EMA model are not trained with gradient descent, but updated as the student changes.
>
> 6. *In VQA downstream task, lines 234, are both the answer decoder and multimodal encoder initialized with the weights of multimodal encoder?*
>
> Yes, both the answer decoder and the multimodal encoder are initialized with the same weights using the pre-trained multimodal encoder.

---

> > ### Comment · Reviewer_z2Jk · 2021-08-25
> > **Post Rebuttal**
> >
> > I have carefully read the authors' response and they address my concerns. I maintain my original rating and agree with the positive reviews from other reviewers.

---

### Official Review · Reviewer_XRzR · 2021-07-16

**Rating:** 7
**Confidence:** 4

**Summary:**

This paper presents an vision-and-language pretraining framework that utilizes momentum distillation to address the issue of training on noisy web data. More specifically, it presents a self-training approach to learn from pseudo-targets that are generated by a momentum model. In contrast to existing approaches which simply use cross-modal attention to reason about concatenated sequences of image regions and words, this work also leverages contrastive learning to learn more effective representations before using cross-modal attention.

**Limitations And Societal Impact:**

Yes, they have.

**Main Review:**

The submission is generally well-written and easy to follow. In particular, the figures are well-drawn and informative, which helps the reader to understand the approach. The related work section is also comprehensive and provides clear distinctions between this work and existing vision-and-language pretraining approaches. The proposed approach is trained using the same pretraining objectives including masked language modeling and image-text matching in existing approaches such as VL-BERT and VisualBERT.

While the concept of knowledge distillation is not new, the application of it to improve the ability of the model to learn from uncurated datasets is interesting. In particular, with noisy text, it helps to improve pretraining by not penalizing the model for producing reasonable outputs that may correspond to the image or text. The provision of theoretical justifications on the benefits of the momentum distillation is also very helpful.

It is not clear how sensitive the proposed approach is to the distillation weight. It would be informative to include an ablation over the distillation weight on a small subset to determine its effect on the final performance. In addition, it is not specified why the text encoder is initialized with the first 6 layers of a pretrained BERT-base model while the multimodal encoder is initialized using the last 6 layers.

The effectiveness of this work is demonstrated via extensive quantitative experiments on multiple downstream tasks, including image-text retrieval and visual question answering, where it outperforms state-of-the-art approaches on all of the metrics. The significance of this work is that the learnt language and visual representations can be used to improve performance for training and evaluating on other downstream tasks.

**Time Spent Reviewing:**

6.5

---

> ### Author Response · Authors · 2021-08-10
> **Response**
>
> We appreciate the reviewer for the positive and insightful feedback. Our response to the reviewer’s questions is as follows.
>
> * *It is not clear how sensitive the proposed approach is to the distillation weight. It would be informative to include an ablation over the distillation weight on a small subset to determine its effect on the final performance.*
>
> We performed an ablation study over the distillation weight $\alpha$ on a subset of pre-training data, where we varied $\alpha$ from 0.2 to 0.6 with a step size of 0.1. Our results show that $\alpha=0.3,0.4,0.5$ yield similar performance, with $\alpha=0.4$ being slightly better than others. We appreciate the question and will include this ablation study in the paper.
>
> * *It is not specified why the text encoder is initialized with the first 6 layers of a pretrained BERT-base model while the multimodal encoder is initialized using the last 6 layers.*
>
> We made a straightforward design choice where the text encoder and the multimodal encoder have the same number of layers, hence each of them inherits 6 layers from a BERT-base (12 layer) model. We leave it as future work to explore other architectural designs.

---

> > ### Comment · Reviewer_XRzR · 2021-08-22
> > **Reply to authors' response**
> >
> > Thank you for providing the conclusions from the ablation study over the distillation weight and the motivation behind the architecture design. This papers provides an interesting approach to use knowledge distillation to learn visual and language representations. Its effectiveness is demonstrated by extensive quantitative experiments.  After reading the author's response, I agree with the positive points raised by other reviewers and keep my initial rating of accept.

---

### Official Review · Reviewer_TH9r · 2021-07-16

**Rating:** 7
**Confidence:** 4

**Summary:**

This paper introduces a method  to perform text-image matching and demonstrate that  it can be used to solve several language-vision downstream tasks  ranging from image-text retrieval, VQA and national language for visual reasoning among others.

* Given an  image and a textual input visual and  textual embedding are obtained  that are then fed  to a multimodal encoder with  cross-attention layers.
* The first key idea of this paper lies in the image-text contrastive loss employed between the image/text embeddings before feeding them to the multimodal  decoder with the intuition behind that being that forcing these embeddings to align before fusing them through the multi-modal encoder results into better representations for downstream tasks
* The second key idea is in the usage of  a momentum model that generates pseudo-target labels by keeping a moving average that are then used for supervision at training time.

An in-depth discussion is provided:
1) for each of the losses used,
2) on the fact that the proposed approach maximizes a lower bound on the mutual information between different “views” of an image-text pair

Extensive Results for several tasks and datasets are provided and strong results are reported. The qualitative results demonstrating where the learned attention is focusing for each downstream task are also quite insightful

**Limitations And Societal Impact:**

The authors have devoted the last 4 lines of the paper to discuss potential limitations/challenges that one should address before deploying this method in real-life applications. There's a large discussion going on these days in the research community in the impact of training large language models and tasks like image-text matching where the visual domain comes in (especially in the context of raw web data) requires additional and careful consideration on what's used, how it's used, how owns the text/img data etc. I think the authors have done a sufficient job in this context but I think a little bit more of an in-depth discussion here would be beneficial.

**Main Review:**

* Well-written, well-motivated, easy to follow and understand what's happening and why the design choices (architecturally, and loss-wise) are done the way they're done. I find the method sufficiently novel in the context of "aligning before fusing" with the contrastive loss. The results reported  across all tasks are strong and SOTA.

* As much as I liked reading the paper and found its results strong and mostly convincing I think it would be a fair question to ask the authors what they think the key novelty of this work is and maybe  discuss it explicitly. The reason I'm mentioning this is that one could argue that it's a combination of existing losses and modules (cross attention, momentum distillation) that are well-put together but at the end of the day there's not an important contribution there.

* Why does the momentum distillation have such a small impact across all downstream tasks (Table 1). I could see why it intuitively makes sense (instead of single positive/negative label have a list of possible pseudo-gt ones) but in practice it seems not super useful?

* While the limitations in terms of  societal impact is indeed provided in Sec. 7, there's no discussion in terms of weaknesses of the proposed approach. For example how fragile is the method in the pseudo-targets generated and what happens if they're consistently off. I'd encourage the authors to provide some additional details and insights here.


## Post Rebuttal

I'd like to thank the authors for taking the time to answer my (and the other reviewers') questions and for providing clarifications on their novelty and some experimental results. Overall I think this is a good paper with an interesting method along with a very solid experimental investigation. I'm happy to increase my score to "7: Good paper, accept".


**Time Spent Reviewing:**

3.5

---

> ### Author Response · Authors · 2021-08-10
> **Response**
>
> We appreciate the reviewer for the positive and insightful feedback. Our response is as follows.
>
> * *As much as I liked reading the paper and found its results strong and mostly convincing I think it would be a fair question to ask the authors what they think the key novelty of this work is and maybe discuss it explicitly. One could argue that it's a combination of existing losses and modules (cross attention, momentum distillation) that are well-put together.*
>
> Two of our key novelties are well-highlighted by the reviewer; these are: (1) image-text contrastive loss to align the unimodal embeddings before fusion, and (2) momentum distillation to generate pseudo-targets. Reviewer z2jK believes our method to be “significantly novel and addresses the limitations of existing work”. Reviewer 16yc considers *align before fuse* to be “new”. Reviewer 3hiK comments on our momentum distillation as “really interesting, and novel". Our paper also makes a third key contribution that theoretically justifies (1) and (2) in a unified framework: we propose a mutual information maximization framework for vision-language pre-training, which Reviewer XRzR finds to be “very helpful”.
>
> Although mutual information for self-supervised learning has been well studied, our paper proposes a novel formulation of “views” for an image-text pair that allows us to unify several losses under the same framework. Formally, we define different views as “unimodal or multimodal data points that have the same semantic meaning but are different in surface form”. View generation can be achieved by either taking non-overlapping and partial information from an image-text pair (i.e., ITC and MLM), or by mining from the dataset (i.e., momentum distillation). We hope that our formulation can inspire new methods for view generation in the field of multimodal learning.
>
> * *Why does the momentum distillation have such a small impact across all downstream tasks (Table 1)?*
>
> In row 4-6 of Table 1, we **incrementally** apply momentum distillation to ITC, MLM and the downstream tasks. In order to see the full effect of momentum distillation, we need to compare row 3 (w/o momentum distillation) to row 6 (w/ momentum distillation), where a significant boost can be observed across all tasks. We appreciate the question and will improve the paper’s clarity.
>
> * *There's no discussion in terms of weaknesses of the proposed approach. For example how fragile is the method in the pseudo-targets generated and what happens if they're consistently off.*
>
> Our method indeed puts requirements on the quality of the pseudo-targets. The distillation loss would be harmful if the pseudo-targets are randomly generated instead of produced by the exponential-moving-average (EMA) model. We have also performed experiments using a pre-trained teacher model to generate pseudo-targets, which leads to further improvement compared to the EMA teacher, but at the cost of additional training time.

---

### Official Review · Reviewer_3hiK · 2021-08-04

**Rating:** 9
**Confidence:** 5

**Summary:**

This paper proposes a new vision-language representation learning framework, ALBEF, which uses image-text contrastive (ITC), masked language modeling (MLM), and image-text matching (ITM) losses with momentum distillation to learn SOTA representations. The ITC and MLM losses are adapted to be convex combinations of the original loss and the KL divergence between the predicted probability distribution (of the image or text or masked tokens) and the soft pseudo targets obtained from the momentum model.

The pipeline is evaluated on image-text retrieval, visual entailment, NLVR, VQA, and grounding and shows improved performance on all tasks compared to prior work, and with fewer training samples in comparison to several prior methods. Qualitative visualizations also verify the quality of the learned representations, with convincing heat maps suggesting well grounded, semantically aligned embeddings.

**Limitations And Societal Impact:**

The authors do not spend significant time in the paper addressing limitations or societal impact, and the writing of the latter is a bit hand-wavy at the end in section 7. However, there are no critical limitations or negative societal impacts that needed to be obviously addressed by the contribution of this paper.

**Main Review:**

This paper was well written and provided extensive quantitative and qualitative evaluation of the representation learning framework. While the individual losses being used were not "original" (contrastive learning has been applied on image-text data before, e.g., CLIP, ALIGN, masked language modeling is commonplace in both language and vision-language representation learning), the used of a momentum model to balance the original losses with a KL divergence term over pseudo target distributions is really interesting, and novel. Additionally, the experimental results are quite convincing, and ALBEF often outperforms prior work with fewer pretraining samples, which is very desirable.

Moreover, the paper clearly states it is trying to address the following problems with prior work:
(1) image and text features residing in independent spaces
(2) the use of object detectors
(3) the overfitting of MLM to noisy text data
And it successfully addresses these points.

An important result of this method is that training with momentum distillation does *not* hurt performance on downstream tasks with clean annotations, unlike prior work where performance can degrade in these settings. It is a significant contribution to use MoD and not degrade performance on clean data.

Authors also provide well-written analysis connecting the losses to maximizing a lower bound on MI. While this is not new information for the ITC loss, it is nice that the authors also connect the MLM loss to this setting. And for readers less familiar with infoNCE prior work, it provides a thorough description.

Questions:
- The ablations in Table 1 verify that using both ITC and ITM (without and with hard negative mining) improves performance over MLM + ITM alone. I find this kind of surprising- why would ITM be complementary to ITC? They seem like very similar losses and contrastive losses already weigh hard negatives more than other samples, so I am wondering if there is any intuition to why the combination of both does better or how ITM inherently differs from ITC.
- How was the alpha chosen for the MoD losses?

**Time Spent Reviewing:**

6 hours

---

> ### Author Response · Authors · 2021-08-10
> **Response**
>
> We appreciate the reviewer for the positive and insightful feedback. We are encouraged to see that our work is recognized as novel and important. Our response to the reviewer’s questions is as follows.
>
> * *Why would ITM be complementary to ITC? I am wondering if there is any intuition to why the combination of both does better or how ITM inherently differs from ITC.*
>
> The objective of ITM is indeed similar to that of ITC. However, ITC operates on the unimodal representations, whereas ITM operates on the multimodal representation. Therefore, ITM enables the model to better learn image-text interactions. As shown in Figure 6, ITM allows the multimodal encoder to capture the mapping between words and image regions through cross-modality attention.
>
> * *How was the alpha chosen for the MoD losses?*
>
> We performed an ablation study over the distillation weight $\alpha$ on a subset of pre-training data, where we varied $\alpha$ from 0.2 to 0.6 with a step size of 0.1. Our results show that $\alpha=0.3,0.4,0.5$ yield similar performance, with $\alpha=0.4$ being slightly better than others. We appreciate the question and will include this ablation study in the paper.

---

> > ### Comment · Reviewer_3hiK · 2021-08-25
> > **Post Rebuttal**
> >
> > Thank you for your response to my questions. With the author response and other positive reviews, I will be leaving my initial rating as is.

---

### Decision · Program_Chairs · 2021-09-27

**Decision:**

Accept (Spotlight)

**Comment:**

This submission proposes to tackle vision-language tasks by combining a contrastive loss used to first “align” the vision & language representations before “fusing” them via a cross-attention model trained via masked language modeling and image-text matching losses.

Reviewers were generally in agreement that this is a strong and well-written submission, proposing a well-motivated approach with convincing state-of-the-art quantitative and qualitative results.  While the overall objective and training procedure is rather complex, each component is ablated and demonstrated to be necessary for the method’s strong performance.

Some reviewers raised a few questions and concerns about missing ablations and design decisions. These questions were addressed satisfactorily by the authors’ responses. I strongly encourage the authors to incorporate the feedback and their provided answers into the camera-ready version of the submission where appropriate.

Given these strengths and especially given the recent interest in the area of vision-language representation and transfer learning, I recommend the submission for a spotlight presentation at NeurIPS.